# Performance Analysis of Wavelength Division Multiplexing-Based Passive Optical Network Protection Schemes by Means of the Network Availability Evaluator

Rastislav Róka 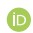

Faculty of Electrical Engineering and Information Technology, Slovak University of Technology, Ilkovičova 3, 812 19 Bratislava, Slovakia; rastislav.roka@stuba.sk

**Abstract:** This paper is focused on the performance analysis of protection mechanisms utilized in common wavelength division multiplexing-based passive optical networks. The main aim of the proposed research is providing an option of comparing different traffic protection scenarios for advanced optical network designs, evaluating their possible realizations from a viewpoint of the total optical power budget and analyzing their network availabilities with accommodation to concrete optical access infrastructures. First, a short basic classification of passive optical network architectures utilizing advanced wavelength division multiplexing techniques is introduced. Second, considered presumptive protection scenarios deployed to this kind of passive optical networks involved in the performance analysis are presented. For the performance analysis, corresponding reliability diagrams, relations and formulas used for the calculation of the total network availability are prepared for any specific scenario. In addition, the optical power budget is considered and subsequently evaluated for each protection scheme. For evaluating possible migration scenarios related to considered protection schemes and for comparing protection possibilities of various passive optical networks, a realization of the appropriate simulation tool must be executed. The simulation program utilizes specific parameters of particular optical components utilized in a selected protection scheme and presents its resultant network availability. Values of considered specific parameters can be changed according to the well-known data resources or specific network operator's data. Subsequently, relevant simulation results obtained by the Network Availability Evaluator that can provide insights into relationships between a number of protection schemes are used for the performance analysis. Finally, an evaluation of the total network availability for considered traffic protection scenarios utilized in passive optical networks with advanced wavelength division multiplexing is included.

**Keywords:** passive optical networks; wavelength division multiplexing; traffic protection schemes; reliability block diagrams; power budget; Network Availability Evaluator; performance analysis

## 1. Introduction

A main concept of Time Division Multiplexing-based Passive Optical Networks (TDM–PON) is a utilization of one shared Optical Line Terminal (OLT) transceiver for communicating with $N$ distant Optical Network Units (ONU) using *1:N* passive optical Power Splitter (PS) elements in the Remote Node (RN) location. All ONU units must by scheduled by means of dedicated time slots controlled by the OLT module. By this way, a number of requested optical components is minimized; however, a performance penalization is occurring at the optical power budget. A disadvantage of this network type is also the Time Division Multiple Access (TDMA) technique that requires a dynamic bandwidth allocation and some services substantial for the latency and the delay do not support it. Above-mentioned problems are not incorporated into future Wavelength Division Multiplexing-based Passive Optical Networks (WDM-PON) principles. In the WDM-PON network, the Arrayed Waveguide Gratings (AWG) element is used instead of the power

splitter, and its loss is much lower. Moreover, a receiver bandwidth can be adjusted to a specific ONU data rate. An advantage of this network type is a direct connectivity between end points and therefore there is no problem with a quality of service because each user is not constrained by other users sharing the same optical infrastructure [1]. Frequently used acronyms and terms are presented in Table A1.

The evolution of passive optical networks is oriented on larger geographical areas, higher numbers of subscribers, and higher bandwidth per subscriber. In [2], a comparison study of active and passive optical access networks utilizing the Point-to-MultiPoint (P2MP) topology is presented without any protection scheme. A possible migration scenario to the next generation of hybrid passive optical access networks is presented in [3] also with a proposal for a self-restored architecture including the classical point-to-multipoint physical topology. In [4], a comparison of power budget analyses for single- and dual feeder fiber architectures of passive optical networks is presented. By means of power budget and transmission performances, a Hybrid Passive Optical Network (HPON) mechanism is analyzed in [5]. In [6], an effect of only fiber duplications on a protection feasibility of passive optical networks is analyzed. In [7], a wavelength reusable WDM-PON self-protected scheme is proposed with protective interconnection fibers used for achieving a whole distribution protection where each ONU is connecting with a ring. Protection mechanisms that can be applied for the whole long-reach passive optical networks are proposed in [8]. In [9], traffic protection schemes considered for these HPON networks are evaluated using the HPON Network Configurator [10–12]. Based on the used network topology, results of this evaluation are presented together with a recommendation that the hybrid and/or WDM-PON network deployment can be done with optimized protection mechanisms for a significant decrease in capital investments. It is profitable to keep at one's disposal an adequate number of reserve optical fibers and/or to provide protection mechanisms at the WDM-PON network deployment. WDM-based networks are considered with a protection and restoration for upgrading current optical access networks in [13]. For WDM-PON networks, a broadcast transmission method is proposed and experimentally implemented without a protection in [14]. In [15], a design and optimization of the hybrid WDM/TDM-PON network based on the ring-tree SARDANA architecture is proposed. An optical access network utilizing the Coarse Wavelength Division Multiplexing (CWDM) technique is proposed with the dual-rate dual-link architecture and with the P2MP topology in [16]. A WDM-PON architecture with the P2MP topology and with alternating optical fiber and optical wireless paths is expressed with a fiber fault protection in [17]. In [18], architectural configuration and operation principles for the metro-access network based on a ring architecture are presented and described together with protection mechanisms for different fiber failure scenarios. Various WDM-PON architectures utilizing only passive components in remote nodes are analyzed from a viewpoint of with the traffic protection securing in [19].

A motivation to this contribution consists of two aspects. First, there exist many scientific papers and research contributions devoted to WDM-PON network designs with various architectures, topologies and components. Only a small part of them consider a practical managerial significance along with practical applications. Thus, they arouse no concern from network operators. In some cases, it is obvious that presented network designs will never be realized and implemented. Second, many network operators devote themselves to a transition to advanced WDM-PON networks. For careful and responsible reviewing and considering diverse approaches, they are coming out above all from own experiences with real customer traffic provisioning in their passive optical infrastructure. They rarely consider other various variants advantageous from long-term aspects for reasons of missing scientific works with qualified examinations of a real WDM-PON network implementation and deployment together with protection scheme mechanisms. Considering both aspects, we decide to create such skillful, flexible and adaptable software tool that can accommodate network operators' demands and can realize a scientific base necessary for qualified and professional decisions associated with a development and

transition to next-generation passive optical networks utilizing advanced wavelength division multiplexing techniques. By this way, we want to provide an option of comparing different traffic protection scenarios for advanced WDM-PON network designs, evaluating their possible realizations from a viewpoint of the total optical power budget and analyzing their network availabilities with accommodation to concrete optical access infrastructures with specific set of optical components, parameters and values.

## 2. Materials and Methods

A network survivability and protection schemes are really important and necessary in advanced optical networks. From a viewpoint of network operators, a network availability is one of the most important factors that can determine a reliability and survivability of selected traffic protection schemes utilized in passive optical networks. Therefore, our performance analysis is based on the Mean Time to Repair (MTTR) and the Mean Time Between Failure (MTBF) parameters that are often and regularly used for evaluation of passive optical networks in operation and that can be used for proving the feasibility of selected traffic protection schemes [20]. This contribution presents the second step of the research work focused on developing of future passive optical networks with new relevant technologies and related traffic protection schemes. Previous work [20] was devoted to the performance analysis of TDM-PON protection schemes by using the PON Network Availability Evaluator. Considering a moderate upgrade to hybrid passive optical networks, traffic protection schemes play an increasing role simultaneously in a development of modern TDM-PON networks. The best network availability can be reached by the dual-parented Type C protection scheme where all components are redundant. It is an advantageous solution, however, with excessive cost demands. The second-best network availability can be reached by the dual-parented Type B protection scheme. In both cases, many reserve components are utilized, above all the most important OLT terminal. Implicitly, advanced traffic protection schemes for more reliable signal transmission and for the higher-level protection safety in next-generation passive optical networks will be considered. After this work, we receive a decision to expand our simulation tool also for advanced WDM-PON protection schemes.

In [21], a comprehensive review of various aspects for WDM-PON technologies is presented. This review includes network architectures, devices, protocols and services proposed for the WDM-PON network deployment. In [22], an efficient resource management approach supporting dynamic bandwidth allocation and sharing downstream wavelengths is proposed for ring WDM-PON architectures. In [23], the ring-based WDM-PON access network with a dual-ring architecture is investigated for providing Rayleigh backscattering noise mitigation and fiber-fault protection. Traffic protection and restoration schemes for advanced hybrid passive optical networks are considered in [24,25]. In [7], a whole optical distribution network protection using protective interconnection fibers is achieved for the wavelength reusable WDM-PON scheme. In [26], different protection schemes for next generation optical access networks are compared based on an assessment methodology. Depending on requirements from the network operator, an appropriate protection scheme can be recommended for both a greenfield and a brownfield scenario based on the targeted network reliability performance. In [27], a physical layer fault management and a protection system for next generation passive optical networks is considered based on a passive optical coding. This approach is oriented on ring-based long-reach passive optical networks. In [28], an analytical model for estimating a network availability under multiple-link failure scenarios is developed for WDM-networks with shared-link connections. For novel hybrid network topologies considered for WDM-PON network designs with traffic protection provision, a selection and optimization of the wavelength grouping on the physical layer can bring other improvements for effective practical utilization of future passive optical networks [29]. In future next-generation passive optical networks, a deployment of the Dense Wavelength Division Multiplexing (DWDM) technique is also expected in a short time. In [30], possibilities for selection of wavelength channels based on

executed performance analyses of the FWM influence are presented for practical utilization in the DWDM communication system.

This research work is focused on developing of future WDM-PON networks with new relevant technologies and related traffic protection schemes and on their performance analysis. The major contributions are listed below:

- Summarize possible and expected architectures of advanced WDM-PON networks;
- Create and make considered presumptive scenarios for protection schemes in advanced WDM-PON networks. By reason of missing such traffic protection and restoration schemes in most of research contributions and scientific papers with WDM-PON network designs, we must propose, design and integrate traffic protection approaches for all possibilities of WDM-PON network architecture including also new protection schemes and mechanisms not considered and presented till now;
- Realize corresponding reliability block diagrams suitable for network availability considerations for protection schemes in advanced WDM-PON networks;
- Execute the performance analysis of WDM-PON protection schemes from a viewpoint of their network availability. For this aim, it is necessary to create, apply, and adjust general relations and formulas for calculation of the total network availability in specific protection scenarios;
- Because the optical power budget is a key parameter for real implementation of any WDM-PON network design, power budget considerations must be evaluated before the performance analysis of WDM-PON protection schemes from a viewpoint of their network availability.
- Enable accommodation and re-adjustment of protection schemes according to network operators' demands from a viewpoint of utilized optical components, their parameters and operating traffic parameters associated with network operators' passive optical network infrastructures.
- Design, create and realize a software tool that will incorporate all necessary relationships between optical components, their relevant parameters and values corresponding to reality and will evaluate a network availability according to network operators' demands.

In this contribution, an importance and possible deployment scenarios of WDM-PON networks together with a network survivability and protection schemes are introduced. In addition, the aim of performance analyzing WDM-PON protection schemes is mentioned. The next sections are organized as follows: in Section 2, a basic classification of WDM-PON network architectures based on the topology is presented. Presumptive scenarios for P2MP and RING protection schemes involved in the performance analysis of WDM-PON networks with corresponding reliability diagrams are presented in Section 3. Subsequently, their network availabilities are considered by the use of appropriate formulas and parameters. Because the power budget in passive optical networks is considered as an important standpoint, WDM-PON protection schemes are considered from this viewpoint and compared based on power budget requirements in addition. For analyzing network availabilities of various possible traffic protection schemes, a modification and extension of the PON Network Availability Evaluator is prepared and created (Section 4). In Section 5, considered protection schemes utilized in WDM-PON networks are evaluated and compared based on their total network availabilities. Moreover, a comparison of TDM-PON and WDM-PON network availabilities is included.

## 3. WDM-PON Network Architectures

The WDM-PON network offers a scalability by supporting more wavelengths on the same optical fiber within its infrastructure and a transparency regarding the channel data rate and a power budget without passive optical losses. For fulfillment, the request for a flexibility in the bandwidth and a number of users, optical components and equipment must be economically acceptable for various architecture types [21]. WDM-PON network architectures can be classified according to the topology as follows:

-      a Point-To-MultiPoint (P2MP) architecture arising from the TDM-PON topology,
-      a RING architecture.

### 3.1. The P2MP Architecture of the WDM-PON Network

For creating this P2MP network architecture, a self-contained wavelength channel from the OLT terminal to each ONU unit must be realized for both downstream and upstream directions. By this, a point-to-point connection is created between OLT and ONU equipment, where each ONU unit can be working on the data rate up to total bit rate of the dedicated wavelength channel. Moreover, different wavelength channels can be working on different bit rates supporting various service variations [21].

### 3.2. The RING Architecture of the Access WDM-PON Network

This RING access architecture connects ONU units in a ring topology where the OLT terminal is connected to this ring through bidirectional optical connections [22,23]. The OLT and ONU equipment must be changed for transmitting and receiving optical signals in downstream and upstream directions. Proposed RING access architecture can support also a re-modulation of the optical signals for increasing a transmission efficiency of dedicated wavelength channels. This architecture offers an advantage of the flexible wavelength allocation where more wavelength channels can be assigned to one ONU unit for increasing the transmission capacity and for the better network scalability.

### 3.3. The RING Architecture of the Metro-Access WDM-PON Network

This RING metro-access architecture is sometimes called the Long-Reach (LR) passive optical network. It is created by the WDM ring with involved passive AWG elements. The specific AWG element is subsequently directed towards the TDM access branch. By using Erbium Doped Fiber Amplifiers (EDFA), two L- and C- bands can be utilized in the optical fiber. At creating this RING metro-access architecture, various splitting ratios can be selected for optical power splitters and a connection of two TDM branches to the one WDM RN with the AWG element is possible.

The ring metro subnetwork has modular possibilities for connecting a large number of users within limits of acceptable attenuation values. It is reached by erbium doped fibers remotely pumped from the OLT terminal. Possible values of the power budget can be adjusted according to fiber lengths and power characteristics of lasers. The access branch subnetwork is created by the TDM-PON network, where the Distribution Fiber (DF) connected to the AWG element is directed to the TDM RN node with the Optical Switch (OS) element. Thereafter, it is splitting up to ONU units through the terminal Drop Distribution Fiber (DDF) [24].

## 4. The Deployment of WDM-PON Protection Schemes

Nowadays, an expansion of WDM-PON networks is arising. Despite their high capital investments, a network resiliency must be increased at a sufficiently high level. Various network architectures are prospective for future WDM-PON networks depending on a way of origin—a greenfield deployment from TDM-PON networks or a brownfield deployment from HPON networks. Considering a moderate upgrade from TDM-PON networks to hybrid passive optical networks, traffic protection schemes (Table 1) simultaneously play an increasing role in the development of modern WDM-PON networks.

**Table 1.** Summary of considered presumptive scenarios for WDM-PON protection schemes.

| P2MP Architectures | RING Architectures |
| --- | --- |
| unprotected P2MP network | unprotected RING access network |
| Type B protection | protected RING access network |
| Dual-parented Type B protection | unprotected RING metro-access network |
| Type C protection | protected RING metro-access network |

### 4.1. Considered Presumptive Scenarios for P2MP WDM-PON Protection Schemes

The P2MP protection scheme of the WDM-PON network is realized by duplicating network components, optical fibers, the OLT terminal, the RN node, and ONU units according to the ITU-T standard [25] for the TDM-PON network. The WDM-PON network is considered as one of broadband solutions for securing the system availability at a certain level that is a principal problem for provisioning high-speed continuous data service. For decreasing a large number of optical fibers, the P2MP protection scheme should offer one or more aggregation layers between the OLT and user locations [7].

Based on the performance analysis in [20], the following presumptive scenarios of the P2MP protection scheme are considered for practical utilization by network operators:

- Type B protection is characterized by the Feeder Fiber (FF) protection. The FF fiber shares a connection with all subscribers and, therefore, its failure means an interruption of the service provisioning for a large number of users. This protection scheme is interesting for most of the operators with smaller up to middle sized traffic load due to its simplicity and low costs [26];
- Dual-parented Type B protection duplicates the OLT equipment along with the FF fiber. In the WDM-PON network, more subscribers are connected to one OLT terminal by using the FF fiber than in the TDM-PON network. As a consequence, failures of the OLT and FF fiber have a large impact on the network reliability and interrupt the service provisioning for larger number of users. Therefore, it is important to protect them [26];
- Type C protection represents a dedicated path protection using a duplication of all network components except the OLT terminal. Components for enlarging the network range are optional. They are used if this protection scheme is utilized for rural areas. A duplication provides a high level of the network availability; however, it requires high capital investments [26].

### 4.2. Considered Presumptive Scenarios for RING WDM-PON Protection Schemes

The RING protection scheme of the WDM-PON network is considered for the protection provisioning in an effective way in terms of costs. Advanced users demand a reliable connection, and it is expected that network operators will provide a continual access to telecommunication network services. Thus, a network reliability securing by implementation of the protection scheme is a substantial request. Optical fibers in a ring topology demand higher attention for failure-less signal transmission comparing to other network topologies. A ring topology decreases costs due to localization and sharing options of the same transmission channels in working and protection optical fibers. This topology can be also used for the TDM-PON or hybrid PON networks [19,23].

For practical utilization by network operators, the following presumptive scenarios of the RING protection scheme are considered:

- The protected access network is designed for the fiber failure protection and the Rayleigh backscattering noise suppression. It provides a possibility for improvement the network scalability and also for increasing the channel capacity per ONU unit. In this protection scheme, there exist two optical paths—working and protection. In a case of the fiber failure, optical modules can protect and restore communication channels by the use of appropriate optical switches for immediate traffic protection providing [23];
- The protected metro-access network utilizes a combination of network topologies. A network is composed of duplicated feeder fibers in the ring topology, whereby two synchronous optical switches are used for selecting either the working or the protection mode. This protection scheme does not provide a protection of the distribution network part. An enormous data traffic up to 1 Tbit/s can be processed, an extra high number of users can be supported and a large geographical area can be covered in this protection scheme. Any failure on the physical layer can cause a huge data damage

affecting end users; therefore, an effective control system for detecting and protecting of failures must be implemented. This protection scheme ensures the high network availability with lower costs comparing to other protection schemes considered for ring network topologies [27].

### 4.3. Reliability Diagrams for WDM-PON Protection Schemes

The network reliability analysis is based on Reliability Block Diagram (RBD) representations due to their significant advantages including an accuracy, flexibility, simplicity and visual impact [3,6]. They allow a determination the total network availability for a given protection scheme. In the RBD scheme, each component is included as a simple block in a correlation with adjacent blocks. Connections in a given protection scheme illustrate unprotected (in series) and/or protected (parallel) components.

For better representation, RBD block schemes are realized for considered WDM-PON protection schemes (Table 1) involved in the performance analysis in Figure 1. RBD diagrams (a)–(d) represent the P2MP WDM-PON protection scenarios according to the ITU-T standard for the TDM-PON network. RBD diagrams (e)–(h) specify RING WDM-PON protection scenarios.

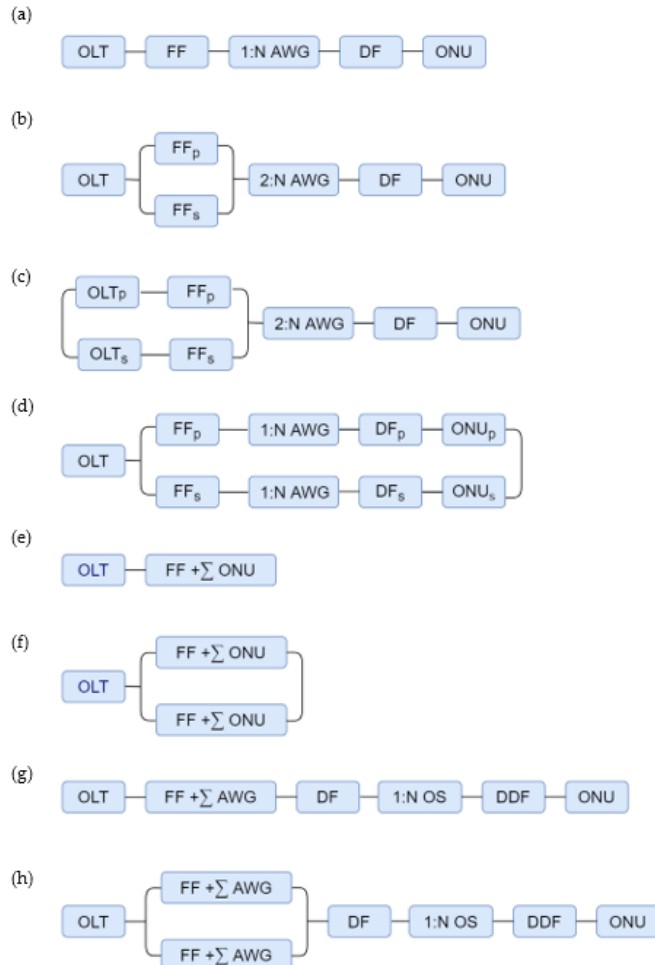

**Figure 1.** RBD diagrams for WDM-PON protection schemes (**a**) the unprotected P2MP network, (**b**) Type B protection, (**c**) Dual-parented Type B protection, (**d**) Type C protection, (**e**) the unprotected RING access network, (**f**) the protected RING access network, (**g**) the unprotected RING metro-access network, (**h**) the protected RING metro-access network.

### 4.4. Network Availability Considerations for WDM-PON Protection Schemes

High resilient WDM-PON networks ensuring the high availability are becoming more relevant due to an increasing number of users and demands for broadband network connections. The network availability is also an important parameter for network operators because it can ensure that the actual network configuration satisfies a requested availability level determined by the Service Level Agreement (SLA). In other way, network operators need to modernize their passive optical network for increasing a level of the network availability. Therefore, a focus on the network availability presents a practical interest especially for services with high demands [28].

A performance analysis of the WDM-PON network availability creates an important characteristic of the functionality for various protection schemes. The WDM-PON network availability represents a probability that all network connections are working properly in any time. The connection availability can be estimated on the basis of failure characteristics for all optical components involved in protection schemes. The connection fails if at least one component is mistaken [28].

For the $n$ components connected in series, the WDM-PON network availability is the same as in the case of the TDM-PON network availability [20]. However, the WDM-PON network availability also includes components connected in parallel [8]. For this case, the total network availability $A_{parallel}$ is presented as follows:

$$A_{parallel} = 1 - \prod_{i=1}^{n} U_{comp_i} \tag{1}$$

where the unavailability parameter $U_{comp}$ represents a failure probability of each component in RBD diagrams determined mathematically using Failure In Time (FIT), MTTR and MTBF parameters defined in [25], and $n$ represents a total number of components in the passive optical network. A deployment and operational situation of the real network infrastructure differs a lot and is depending on a specific network operator and/or area.

Decreasing the time for repairing optical fibers and other network components markedly affects the network availability. Equations for the network availability calculation differ based on P2MP and RING protection schemes due to different topologies and utilized network components. Hereby, the unavailability for each WDM-PON network can be determined. For evaluation of the total network availability for each considered WDM-PON protection scheme, RBD block diagrams, due to their simplicity and understanding given protection schemes, are used as presented in Figure 1.

The total availability of the unprotected P2MP WDM-PON network (Figure 1a) is calculated by using

$$A_{P2MP\_unprotected} = 1 - (U_{OLT} + U_{FF} + U_{AWG} + U_{DF} + U_{ONU}) \tag{2}$$

with unavailability parameters $U_x$ of the OLT module, the FF fiber, the AWG element, the DF fiber and the ONU unit for specific lengths of feeder and distribution fibers [3].

A calculation of the network availability for P2MP protection scenarios furthermore includes a parameter of the network restoration time [20,25]. The restoration time is a time reserved for switching to redundant components and for traffic restoration. For WDM-PON networks, values of the restoration time are considered in tens of ms.

The total availability of the unprotected RING access WDM-PON network (Figure 1e) is calculated by using

$$A_{RING\,A\_unprotected} = 1 - \left( U_{OLT} + \frac{FF_A}{2} \cdot U_{FF} + \frac{k+1}{2} \cdot U_{ONU} \right) \tag{3}$$

where $FF_A$ is the feeder fiber length in the ring, and $k$ is a number of *ONU* units connected to the ring [27].

The total availability of the unprotected *RING* metro-access WDM-PON network (Figure 1g) is calculated by using

$$A_{RING\ M-A\_unprotectd} = 1 - \left( U_{OLT} + \frac{FF_M}{2} \cdot U_{FF} + \frac{l+1}{2} \cdot U_{AWG} + \frac{DF_A}{2} \cdot U_{DF} + U_{PSC} + U_{ONU} \right) \quad (4)$$

where $FF_M$ is the feeder fiber length in the WDM ring, and $l$ is a number of AWG elements. The TDM branch network involves $DF_A$ representing the length of distribution fibers (DF + DDF), the optical power splitter/combiner, and the $ONU$ units connected to the ring through the appropriate AWG element [27].

　　A calculation of the network availability for RING protection scenarios includes the FF redundancy and thus a parameter of the fiber restoration time is also involved. Moreover, this calculation is affected also by a number of particular optical components considered at the ring architecture design.

*4.5. Power Budget Considerations for WDM-PON Protection Schemes*

　　The evolution of WDM-PON networks directs towards larger coverage of access areas, higher number of subscribers and wider available broadband for services [19]. The Power Budget (PB) in the WDM-PON network is an important parameter for the designing of optical telecommunication networks together with other considered features as a network type, a utilized technology, a network range, a traffic protection, and a mutual interconnection of network components.

　　The available optical budget $PB_{WDM}$ involving losses due to various components used in WDM-PON networks can be mathematically expressed as follows:

$$PB_{WDM} = P_{Tmin} - R_{Smin}\ [\text{dB}] \quad (5)$$

where $P_{Tmin}$ is the minimum transmitted power generated by the OLT laser source launched onto the optical fiber in [dBm], and $R_{Smin}$ presents a minimum average power requested by the ONU receiver in [dBm] [5].

　　The signal attenuation is given as a sum of losses from all components $L_{WDM\ comp}$ as

$$\sum L_{WDM\ comp} = \alpha \cdot FF + \alpha \cdot DF + L_s + m \cdot L_{con} + L_{AWG} + L_{MUX/DEMUX} + SM\ [\text{dB}] \quad (6)$$

where $FF$ presents a length of the feeder fiber in [km], and $DF$ presents a length of the distribution fiber in [km]. In a case of the WDM-PON network, the *(FF + DF)* fiber length is taken from a 20 km to 100 km range. The specific attenuation $\alpha$ gently differs for various utilized wavelengths in the WDM-PON network. For the power budget calculation, it can be considered $\alpha$ = 0.25 [dB/km] at the 1550 nm central wavelength for a standard single-mode fiber. The loss due to splices is $L_s$ = 0.2 [dB], whereby one splice is realized at each 5 km. The loss due to connectors is $L_{con}$ = 1 [dB], whereby one connector is used at 10 km, and $m$ is a number of connectors used. For the AWG element, a provision of the wavelength routing for eight different wavelengths around the 1550 nm central wavelength is supposed. The attenuation of this element is $L_{AWG}$ = 5 [dB]. The losses due to multiplexers and demultiplexers are represented by $L_{MUX/DEMUX}$ = 2 [dB]. The System Margin (SM) allocates a certain amount of power to additional sources of power penalty that may develop during a system lifetime because of a component degradation or other unforeseen events. A system margin is from the 3 to 10 dB range, and it is allocated during a process of network designs [31].

　　Subsequently, the calculated optical power budget $PB_{WDM\ calc}$ in passive optical networks [5] is determined as follows:

$$PB_{WDM\ calc} = PB_{WDM} - \sum L_{WDM\ comp}\ [\text{dB}] \quad (7)$$

　　For correct power budget considerations, both input and output power levels at OLT and ONU modules and simultaneously all kinds of attenuations occurring in the passive optical network should be known [5]. For evaluation of the total network availability for each considered WDM-PON protection scheme, a power budget and a number of

subscribers due to their impacts on functionality and operation of given protection schemes must be necessarily included.

### 4.6. The Evaluation of the Power Budget in WDM-PON Protection Schemes

For the PB calculation, results for the P2MP WDM-PON protection schemes based on specific parameters' values from Table 2 are presented in Figure 2. The highest power budget is determined for the unprotected P2MP network. The Type B protection satisfies requirements for correct evaluation of the power budget. For more protection, the dual-parented Type B or Type C protections can be adequately selected. These protection scenarios are suitable for rural geographical areas up to 33 km fiber length.

**Table 2.** Specific parameters' values for power budget calculations in WDM-PON networks.

| Parameter | P2MP Architectures | RING Architectures |
|---|---|---|
| Transmitted optical power $P_T$ [dBm] | 8 | 10 |
| EDFA gain $G$ [dB] | - | 31 |
| Specific attenuation $\alpha$ [dB/km] | 0.25 | 0.25 |
| Feeder fiber length $FF$ [km] | 25 | 10 |
| Distribution fiber length $DF$ [km] | 8 | 3 |
| Splice loss $L_s$ [dB] | 0.2 | 0.2 |
| Connector loss $L_{con}$ [dB] | 1 | 1 |
| AWG loss $L_{AWG}$ [dB] | 5 | 5 |
| Mux/Demux loss $L_{Mux/Demux}$ [dB] | 2 | - |
| OLT with Mux/Demux loss $L_{OLT\ Mux/Demux}$ [dB] | - | 2 |
| ONU with Mux/Demux loss $L_{ONU\ Mux/Demux}$ [dB] | - | 2 |
| Number of ONU units | - | 10 |
| Number of optical power splitters | - | 16 |
| System margin $SM$ [dB] | 3 | 3 |
| Receiver sensitivity $R_S$ [dBm] | −35 | −28 |

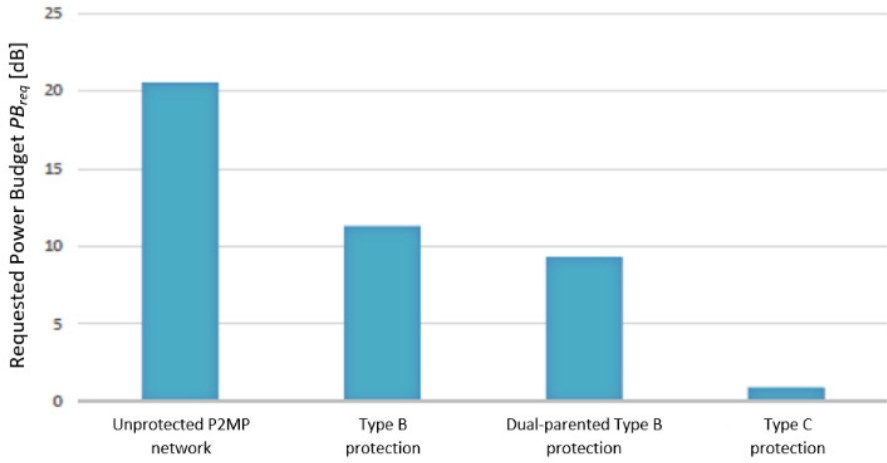

**Figure 2.** The comparison of power budget requirements for P2MP WDM-PON protection schemes.

Upon designing the WDM-PON network with P2MP protection schemes, it is important to accommodate values of specific parameters for acquiring a correct, reliable and convenient optical power budget of the final selected protection scheme.

For the PB calculation, results for the RING WDM-PON protection schemes based on specific parameters' values from Table 2 and another adequate basic data selected for this topology type are presented in Figure 3. The 13 km fiber length is used for covering a metropolitan area with a higher number of users. The highest power budget must be assured for the unprotected metro-access network. However, the protected network architecture is more suitable because this architecture ensures the protection by means of

the FF fiber redundancy, whereby no system breakdown influencing a high number of users is occurring in a case of the fiber failure.

**Figure 3.** The comparison of power budget requirements for RING WDM-PON protection schemes.

Upon designing of the WDM-PON network, an arrangement of specific parameters is also possible for these ring protection schemes. However, the power budget analysis is again necessary for acquiring a correct, reliable, and convenient optical power budget in a given selected protection scheme.

## 5. The PON Network Availability Evaluator

The PON Network Availability Evaluator is a simulation program that allows comparing various traffic protection schemes utilized in passive optical networks from a viewpoint of the network availability. In [20], a performance analysis of TDM-PON protection schemes is presented. For this contribution, the PON Network Availability Evaluator must be modified and extended for obtaining relevant results available for advanced protection schemes. The realized simulation program evaluates real scenarios of the WDM-PON protection schemes based on various specific technical specifications—a protection scheme, a restoration time, values of MTTR and MTBF parameters for utilized optical components. The PON Network Availability Evaluator is created in the Microsoft Excel program environment by means of ActiveX control elements and by utilization of the Visual Basic for Applications (VBA) code. The simulation program allows for configuring specific parameters of traffic protection schemes considered for the WDM-PON networks. The client can intuitively control a program interface and arrange values of requested specific parameters according to real network implementations.

The simulation program is oriented on both-P2MP and RING-WDM-PON protection schemes. For each protection scheme, various presumptive scenarios are considered.

A calculation of the WDM-PON network availability is realized by using of values network presented in Table 3 for each network component. Basic parameters are divided according to the topology, whereby the fiber length is corresponding to the WDM-PON protection scheme. Table 3 summarizes some data related to the network unavailability as a consequence of failures of WDM-PON network components [7].

The simulation program utilizes values of MTTR and MTBF parameters for particular optical components and directly presents a resultant value of the network availability for a selected WDM-PON protection scheme. Each component is changing independently and/or simultaneously with other components depending on the selected protection scheme. Therefore, values of considered specific parameters for the OLT terminal, ONU units, AWG and OS elements, FF and DF fibers can be changed according to the [7] or specific network operator's data, if available.

**Table 3.** Specific parameters' values for components in WDM-PON networks.

| Component | MTBF [H] | | MTTR [h] | FIT | Fiber Length [km] | |
| | P2MP | RING | | | P2MP | RING |
|---|---|---|---|---|---|---|
| OLT | 400,000 | 400,000 | 2 | 256 | - | - |
| FF | 70,175 | 175,438 | 24 | 570/km | 25 | 10 |
| RN | 8,333,333 | 8,333,333 | 6 | 120 | - | - |
| AWG | 5,000,000 | 5,000,000 | 6 | 200 | - | - |
| DF | 219,298 | 584,795 | 24 | 570/km | 8 | 3 |
| ONU | 3,906,250 | 3,906,250 | 6 | 256 | - | - |
| OS | - | 5,000,000 | 6 | 200 | - | - |

*The Main Screen of the PON Network Availability Evaluator*

First, the main screen is presented with default parameters of the P2MP WDM-PON protection scheme for the unprotected network scenario (Figure 4). On the left side, the P2MP topology can be selected and subsequently related protection scenarios are offered. For all protection scenarios, the expected restoration time is necessary to select. The button CALCULATE starts a process for calculating the network availability in [%]. For better understanding, a value of the Mean Down Time (MDT) in [min/year] related to the network unavailability is depicted.

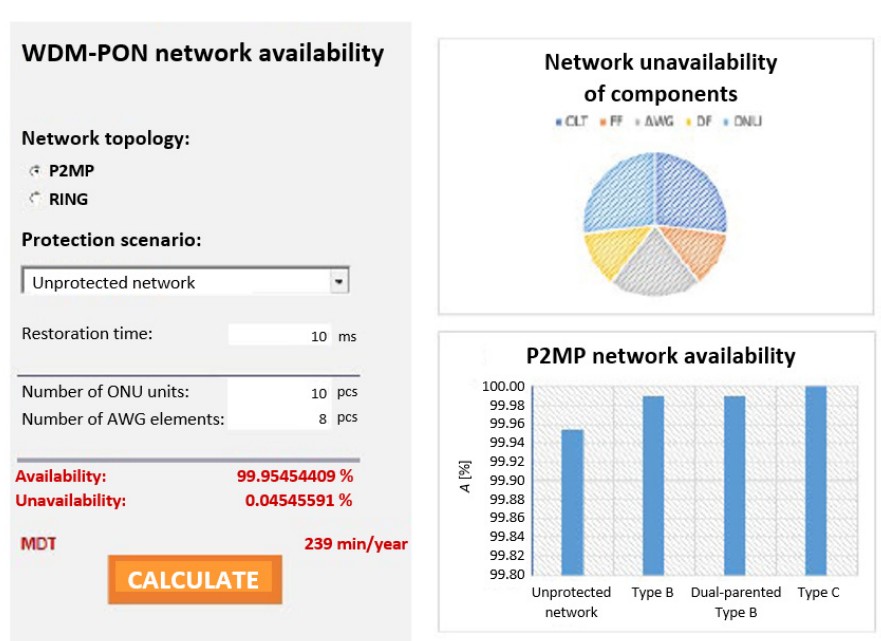

**Figure 4.** The main screen's part for the unprotected P2MP WDM-PON network.

In the middle of the screen, two graphs related to the selected protection scheme are presented. The upper graph charts the network unavailability of components, and the lower graph presents a comparison of the network availability for all protection scenarios offered for the selected network topology. Immediately after selecting the protection scenario and setting options, resultant values of the network availability and unavailability are shown.

On the right side, a WDM-PON network architecture corresponding to the selected P2MP protection scenario is then pictured. For better understanding, an explanation of active working and protection elements and redundant parts in different architectures is embedded.

Next, the main screen with modified parameters of the RING WDM-PON protection scheme for the protected metro-access network scenario—a ring topology, a protected

metro-access network—is displayed (Figure 5). Furthermore, parameters—the restoration time 5 ms and a number of AWG elements is 6—are setting up. After pressing the button CALCULATE, results for the network availability and unavailability and the MDT time are re-calculated and a graph of the network unavailability of components is automatically changed. The change is also expressed by a WDM-PON architecture for the selected protection scenario.

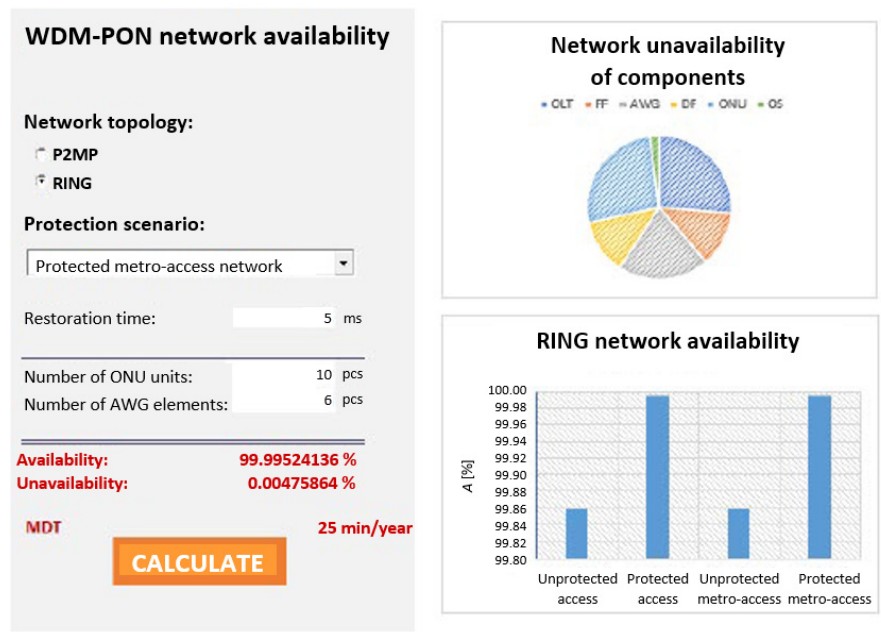

**Figure 5.** The main screen's part for the protected RING metro-access WDM-PON network.

## 6. The Evaluation of the Network Availability for WDM-PON Protection Schemes

Using the PON Network Availability Evaluator, the total network availability for considered scenarios of WDM-PON protection schemes can be evaluated and compared. Resultant network availabilities (Figure 6 and Table 4) are calculated for specific parameters' values from the data resource [7] and for selected restoration times. Possible changes of network availabilities are depending on a type of the architecture, on a specific protection scenario and on the restoration time.

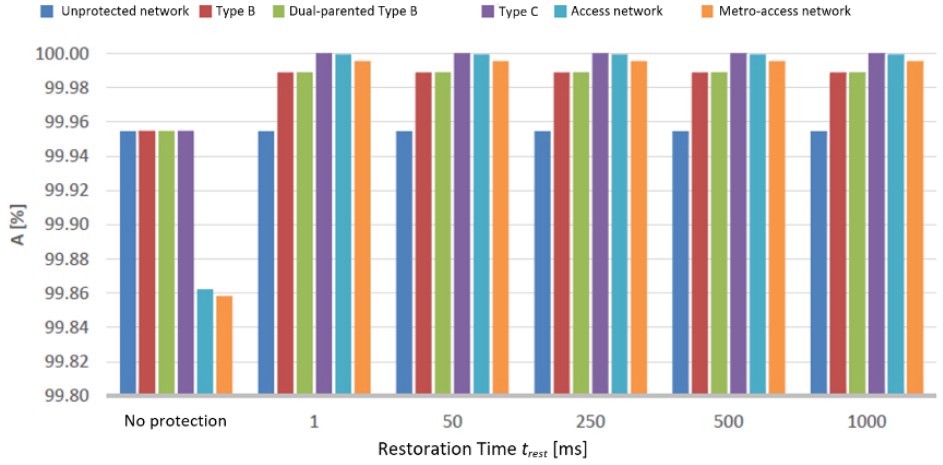

**Figure 6.** The total WDM-PON network availability for selected restoration times.

**Table 4.** The total WDM-PON network availability for selected restoration times.

| | | P2MP | | | RING | |
|---|---|---|---|---|---|---|
| | | **Type B** | **Dual-Parented Type B** | **Type C** | **Access** | **Metro-Access** |
| No protection | | 99.95454409 | 99.95454409 | 99.95454409 | 99.86234142 | 99.85835879 |
| $t_{rest}$ [ms] | 1 | 99.98873240 | 99.98878360 | 99.99994880 | 99.99910400 | 99.99512137 |
| | 50 | 99.98873238 | 99.98878358 | 99.99994877 | 99.99910392 | 99.99512129 |
| | 250 | 99.98873230 | 99.98878350 | 99.99994867 | 99.99910361 | 99.99512097 |
| | 500 | 99.98873220 | 99.98878340 | 99.99994853 | 99.99910321 | 99.99512058 |
| | 1000 | 99.98873200 | 99.98878320 | 99.99994826 | 99.99910242 | 99.99511979 |

*6.1. The Evaluation of the P2MP WDM-PON Protection Scenarios*

The network availability of P2MP WDM-PON protection schemes achieves different values for different protection scenarios. The highest network availability is reached by the Type C protection. Other protection scenarios—Type B and Dual-parented Type B—reach similar results. In Figure 7, the network availability of the dual-parented Type B protection is presented for various protection times from Table 4. As can be seen, this network availability is decreasing with higher restoration times.

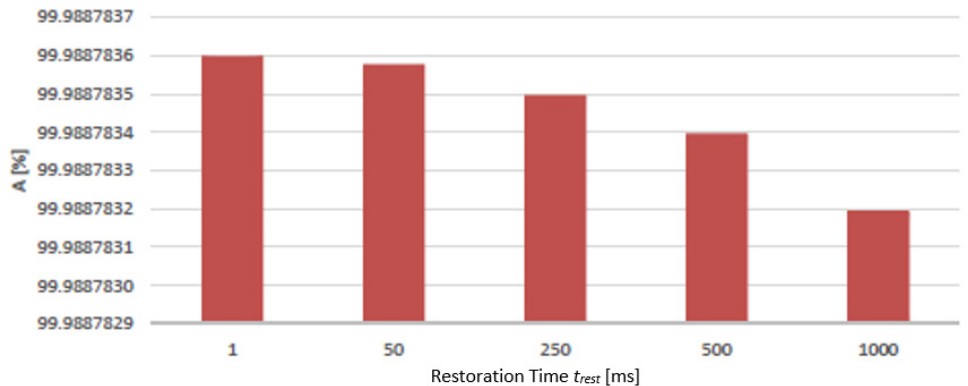

**Figure 7.** The network availability of the dual-parented Type B protection for selected restoration times.

*6.2. The Comparison of the TDM-PON and P2MP WDM-PON Network Availabilities*

For this comparison, the following parameters are considered in the PON Network Availability Evaluator—the restoration time 10 ms, the unprotected network, Type B, the dual-parented Type B and Type C. The network availabilities for TDM-PON and P2MP WDM-PON networks for this restoration time and selected protection scenarios are presented in Figure 8. Particular results are based on values of specific parameters for the TDM-PON network presented in [20] and for the P2MP WDM-PON network presented in [7].

As can be seen, better resultant values of the network availability are reached in the TDM-PON network, where results differ slightly for considered protection scenarios. In a case of the P2MP WDM-PON network, there are significant differences between the unprotected network, Type B and Type C protection scenarios. Because network availabilities for Type B and dual-parented Type B protections are nearly identical, a network realization with the dual-parented Type B protection is more suitable for the WDM-PON practical utilization due to higher protection of several network components. The Type C protection reaches higher network availability, but with only small variance from others. The network availability of P2MP WDM-PON protection scenarios is lower because a more complex system and more complicated elements are necessary for their correct performance and functionalities. Reversely, TDM-PON protection scenarios contain a more simple system with network components consisting of a smaller number of basic elements.

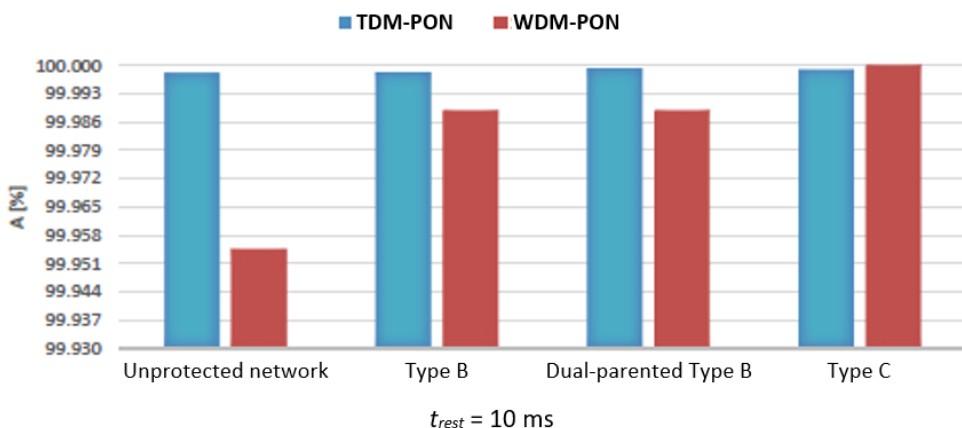

**Figure 8.** The network availability of TDM-PON and P2MP WDM-PON protection scenarios for the selected restoration time.

### 6.3. The Evaluation of the RING WDM-PON Protection Scenarios

In Figure 9, the network availability of the RING WDM-PON protection schemes is presented for different protection scenarios—unprotected and protected access networks and unprotected and protected metro-access networks. For this comparison, the following parameters are considered in the PON Network Availability Evaluator—the restoration time 10 ms, a number of ONU units for the access network is 10, and a number of AWG elements for the metro-access network is 8.

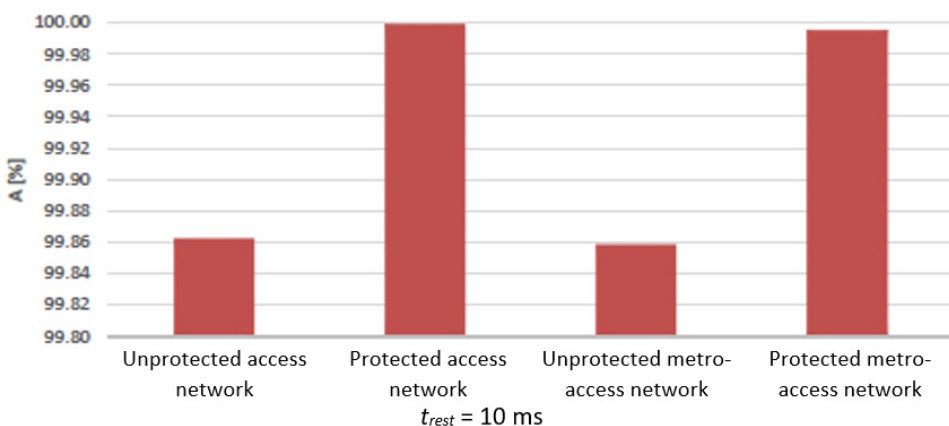

**Figure 9.** The network availability of RING WDM-PON protection scenarios for the selected restoration time.

As can be seen, better resultant values of the network availability are reached in protected network scenarios. The protected access network has higher network availability; however, its variance from the protected metro-access network is weak. Moreover, the protected metro-access network is simultaneously more suitable for the WDM-PON practical utilization due to a higher number of users.

Upon changing the parameters, the network availability of RING WDM-PON protection scenarios for the 10 ms restoration time is presented in Figure 10. We can see that the network availability is decreasing with more network elements. Thus, we can conclude that a number of elements has larger importance than the restoration time in RING WDM-PON protection scenarios. In addition, a fiber length is an important parameter influencing a decrease in the network availability.

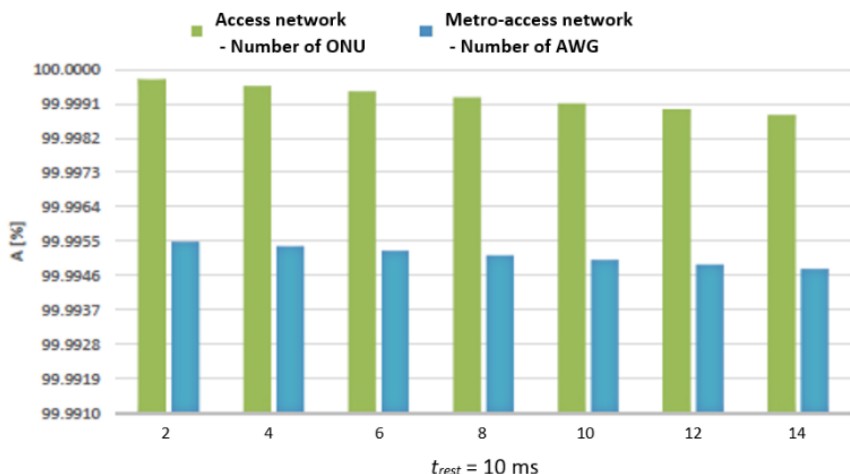

**Figure 10.** The network availability of RING WDM-PON protection scenarios for different numbers of elements.

The protected access network has markedly better results with five places nine availability, when the protected metro-access network availability has only four places nine. However, the protected metro-access network provides traffic for a higher number of users and can be easily modified according to demands of specific network operators.

## 7. Discussion

Nowadays, TDM-PON protection schemes are not expanded in an adequate way. Due to a low number of users, capital investments in TDM-PON networks must be minimized and simultaneously a network resiliency must be increased sufficiently. This situation is changed for advanced WDM-PON networks, where the traffic protection and restoration mechanisms must not only be considered, but also realized.

Depending on a rate of development and projected range of advanced WDM-PON networks, network operators can take into account various traffic protection and restoration scenarios. An important and active part of this aim is a fact if a considered WDM-PON network only accommodates an original TMD-PON network architecture or fully exploits its own potential related to wavelength management features in selected network topologies, eventually with an interconnection with a metropolitan network. For this reason, presented simulation results of the performance analysis for WDM-PON protection schemes can be very useful for their practical applications.

In the future, TDM-PON and WDM-PON protection schemes can be analyzed based on CAPEX and OPEX costs of particular network components. For this purpose, the PON Network Availability Evaluator will be modified, its graphical interface can be adjusted, and new network elements can be added and its extension with another protection schemes can be realized. Moreover, wavelength multiplexing techniques can be implemented into existing protection schemes. Except for standard values of specific parameters, a targeted utilization of data provided by real network operators together with implemented traffic protection schemes will be very interesting.

## 8. Conclusions

In this contribution, possible traffic protection schemes for common wavelength division multiplexing-based passive optical networks are presented, analyzed and evaluated. We realized reliability block diagrams suitable for network availability considerations for WDM-PON protection schemes. We created and adjusted formulas for calculation of the total network availability in specific protection scenarios. Before the performance analysis, we evaluated and analyzed optical power budget considerations for WDM-PON protection schemes. This is a strong recommendation for any other network designs.

For evaluating possible network migration scenarios related to protection schemes and for comparing of protection possibilities of various passive optical networks, we proposed, modified and extended the PON Network Availability Evaluator simulation tool. Comparing to a previous performance analysis of TDM-PON protection schemes, the PON Network Availability Evaluator must be upgraded for obtaining relevant results appropriate for the performance analysis of WDM-PON protection schemes. This program incorporates all relationships and connections between optical components with their relevant parameters and real values according network operators' demands necessary for evaluating a network availability in WDM-PON protection scenarios. Subsequently, we realized the performance analysis by means of the PON Network Availability Evaluator. The performance analysis is based on corresponding reliability block diagrams and optical power budget considerations and on specific parameters used for the calculation of the total network availability for each considered WDM-PON protection scheme.

Based on selected specific parameters, various considered presumptive scenarios of P2MP and/or RING WDM-PON protection schemes can be evaluated and compared from a viewpoint of the network availability. Presented simulation results of the network availability for WDM-PON protection schemes are analyzed and evaluated from a viewpoint of their practical utilization and implementation. The evaluation of the WDM-PON network availability consists of three parts. In the first part, the highest network availability for P2MP protection scenarios is reached by the Type C protection. Other considered protections reach similar results, whereby the worst scenario is an unprotected network. In addition, network availability dependencies on restoration times can be obtained for any considered protection scheme. In the second part, a comparison of original and advanced passive optical networks is possible due to the same P2MP architecture. Better resultant values of the network availability are reached in the TDM-PON network representing a more simple system with a smaller number of simpler network components. For P2MP WDM-PON protection scenarios, the dual-parented Type B protection is suitable for a practical utilization due to higher protection of several network components. In the third part, better resultant values of the network availability are reached in protected network scenarios compared to unprotected networks. For the WDM-PON practical utilization, the protected metro-access network is more suitable due to a higher number of users than the access only network. In addition, network availability dependencies on different numbers of relevant optical elements can be acquired for protections of any network type. By this way, a practical significance of our research along with its practical applications can be presented.

**Funding:** This work was funded by Kultúrna a Edukačná Grantová Agentúra MŠVVaŠ SR, Grant No. KEGA 034STU-4/2021.

**Institutional Review Board Statement:** Not applicable.

**Informed Consent Statement:** Not applicable.

**Data Availability Statement:** Not applicable.

**Conflicts of Interest:** The author declares no conflict of interest. The funders had no role in the design of the study; in the collection, analyses, or interpretation of data; in the writing of the manuscript, or in the decision to publish the results.

## Appendix A

**Table A1.** Summary of frequently used acronyms and terms.

| Acronym | Meaning |
| --- | --- |
| AWG | Arrayed Waveguide Grating |
| CWDM | Coarse Wavelength Division Multiplexing |
| DDF | Drop Distribution Fiber |
| DF | Distribution Fiber |
| DWDM | Dense Wavelength Division Multiplexing |
| EDFA | Erbium Doped Fiber Amplifier |
| FF | Feeder Fiber |
| FIT | Failure In Time |
| HPON | Hybrid Passive Optical Network |
| LR | Long Reach |
| MDT | Mean Down Time |
| MTBF | Mean Time Between Failure |
| MTTR | Mean Time To Repair |
| OLT | Optical Line Terminal |
| ONU | Optical Network Unit |
| OS | Optical Switch |
| P2MP | Point-To-Multi Point |
| PB | Power Budget |
| PON | Passive Optical Network |
| PS | Power Splitter |
| RBD | Reliability Block Diagram |
| RN | Remote Node |
| TDM | Time Division Multiplexing |
| TDMA | Time Division Multiple Access |
| VBA | Visual Basic for Applications |
| WDM | Wavelength Division Multiplexing |

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
