# Peer review of "Performance Analysis of Wavelength Division Multiplexing-Based Passive Optical Network Protection Schemes by Means of the Network Availability Evaluator"

_applsci, doi:10.3390/app12157911_

Round 1
Reviewer 1 Report
The authors present a study which builds on previous work in the field. The authors identify the research as: : a performance analysis of protection mechanisms for "wavelength division multiplexing-based passive optical networks". I found that the manuscript lacks a logical structure and the narrative is unclear. I have comments:
1) The title needs revision to better reflect the subject and topics considered in the paper. Moreover, the title contains acronyms which must be removed in a revised and improved title.
2) The abstract requires revision to better reflect the paper and improve the discussion around the results which are not adequately addressed. Moreover, the abstract should clearly set out the: Why? (the motivation), How? (the methodology), and What? (the results and conclusions drawn). Additionally, all acronyms must be removed from the abstract with acronyms limited to the main body text with a definition on first use.
3) The keywords are adequate bus acronyms must be removed.
4) The Introduction is in reality both an introduction and an overview of related research. The Introduction must be revised in two dedicated sections: (1) an Introduction setting out: (a) the background, (b) the motivation, (c) a brief overview of the research introduced, (d) the claimed contribution clearly set out (I noted comments in the conclusion), and (e) the paper structure. (2) a section considering the related research with a comparative analysis setting out the relative merits of each paper considered along with a comparative analysis.
5) I noted the observation that the current paper builds on previous research. The paper lacks a discussion section where the research (current and previous) is considered in adequate detail. Provision of such a section is essential.
6) The conclusion section requires revision to restrict the contents to closing observations. Currently it contains material that should be located in other sections (e.g., future work and contribution).
7) In all such research there will be open research questions (ORQ) which arise. In this paper such questions must address both the current and previous work. The brief discussion on future work [in the conclusion] is not adequate and fails to consider ORQ and related directions for research in sufficient detail. This is essential and must be provided possibly in the new discussion section.
8) I all such research we must consider the practical managerial significance along with practical application. This is required in a revised version of the paper where practical application of the research is discussed along with illustrative scenarios demonstrating the utility of the research.
9) I noted the extensive use of acronyms in the text and related figures. There should be a list of acronyms the make reading the paper simpler.
10) The figures are in many cases too small and for figures 4 and 5 the formatting is incorrect (column overflow). The formatting of the manuscript requires attention to address such issues and the authors need to consider the captions to improve the descriptions.
In summary, the paper requires extensive revision as noted above to improve the presentation, logical structure, narrative, and practical application. Upon suitable revision the research documented in the manuscript can be better evaluated.
Reviewer 2 Report
This paper studies the WDM-PON Protection Schemes by using PON Network Availability Evaluator.
major concerns:
1. the motivation and contribution of this paper are unclear and should be improved. It is not clear what the author is proposing.
2. The paper organization also needs to be improved to show clearly what the author proposed.
3. Majority of the paper only discusses the calculation of different protection schemes with the PON Network Availability Evaluator, without any novel proposed architecture protection scheme mechanism.
Minor concerns:
1. some Figures are too small to read.
2. please clarify the simulation scenarios, a table would help the reader to understand better.
3. please explain, the protection scheme in more detail. For example, is this protection for the link or ONU/OLT fault.
Reviewer 3 Report
1. The abstract is very ambiguous and does not present the main contributions of the paper. The main aim of the proposed study is missing in the abstract. Authors should also highlight the major contributions of the paper and significant results in the abstract to promote the importance and performance of their proposed approach.
2. I am finding it difficult to figure out the research gap in this paper as well as the novelty of the proposed approach. The introduction section should clarify better and provide the motivation behind this study, research gap, novelty, and contributions of the paper. The introduction should also be updated to provide concise information about the problem definition and scope of the paper.
3. Simulation parameters must be provided.
4. Detailed result analysis is missing.
5. The conclusion section should present a summarization of the main contributions, results, limitations and future research directions.
Author Response
1. The abstract is very ambiguous and does not present the main contributions of the paper. The main aim of the proposed study is missing in the abstract. Authors should also highlight the major contributions of the paper and significant results in the abstract to promote the importance and performance of their proposed approach.
Response: The comment is accepted; the revised abstract contains a main aim and contributions of the paper where the importance and performance of a proposed research is clearly presented. The main contributions consisting of seven separate parts are then specifically listed in the section 2 “Materials and Methods”.
2. I am finding it difficult to figure out the research gap in this paper as well as the novelty of the proposed approach. The introduction section should clarify better and provide the motivation behind this study, research gap, novelty, and contributions of the paper. The introduction should also be updated to provide concise information about the problem definition and scope of the paper.
Response: I would like to thank Reviewer for a mentioned opinion. However, I cannot agree with this comment. The revised manuscript contains new sections and text parts where the research gap as well as the novelty and contributions of this paper is clearly presented. Also, Introduction is extended for better presentation of a research contribution, a problem definition and a scope of the paper.
3. Simulation parameters must be provided.
4. Detailed result analysis is missing.
5. The conclusion section should present a summarization of the main contributions, results, limitations and future research directions.
Response: I would like to thank Reviewer for mentioned comments. However, I cannot agree with these comments. In this work :
- simulation parameters related to the performance analysis are provided in Table 2 and Table 3 related to all WDM-PON protection schemes considered in Table 1. Moreover, additional concrete parameters utilized for presented evaluations are introduced in subsections 4.5 and 4.6.
- detailed result analysis is presented in the section 6 “The evaluation of the network availability for WDM-PON protection schemes” containing three subsections, five figures and one table.
- sections Discussion and Conclusion present main contributions, results, limitations and future research directions related to this paper.
Round 2
Reviewer 1 Report
I have read the revisions cover letter and revised manuscript. In general the revisions address the principal comments expressed in my review. The structure is not ideal (it is somewhat different from the normal) but is acceptable. I do have some recommendations and suggestions:
1) The structure of the introduction would be improved if a clear paper structure was specified with comments pointing to the appendix (acronyms).
2) There are some formatting issues which will be addressed by the editor in the proofing process.
3) The content related to open research questions and future work would benefit from increased detail and discussion as would the practical managerial significance.
In summary, I have made suggested improvements (not obligatory but recommended) which will improve the manuscript. When properly proofed in the usual proofing process the paper may be considered for publication.
Author Response
1) The structure of the introduction would be improved if a clear paper structure was specified with comments pointing to the appendix (acronyms).
Response: I'd like to thank Reviewer for a mentioned comment that makes this paper better. The Introduction is revised and a new comment pointing to the acronyms and terms is added.
2) There are some formatting issues which will be addressed by the editor in the proofing process.
Response: The comment can be accepted. I tried to fulfill any formatting requests; if there are some issues, I hope that they will be addressed in the proofing process.
3) The content related to open research questions and future work would benefit from increased detail and discussion as would the practical managerial significance.
Response: I'd like to thank Reviewer for a mentioned comment that makes this paper better. This comment is accurate and substantial and helped me improve this manuscript more comprehensively in more detail. New revised parts related to the practical managerial significance can be found in sections Introduction, Discussion and Conclusion. They are presented in a way that is not oriented to one specific case, but in a sufficiently universal approach applicable to various network operators’ considerations.
In summary, I have made suggested improvements (not obligatory but recommended) which will improve the manuscript. When properly proofed in the usual proofing process the paper may be considered for publication.
I am highly pleased to have received your comments.